# High-resolution temporal profiling of *E. coli* transcriptional response

Arianna Miano [1] ✉, Kevin Rychel [1], Andrew Lezia[1], Anand Sastry[1], Bernhard Palsson[1,2] & Jeff Hasty[1,3,4]

Understanding how cells dynamically adapt to their environment is a primary focus of biology research. Temporal information about cellular behavior is often limited by both small numbers of data time-points and the methods used to analyze this data. Here, we apply unsupervised machine learning to a data set containing the activity of 1805 native promoters in E. coli measured every 10 minutes in a high-throughput microfluidic device via fluorescence time-lapse microscopy. Specifically, this data set reveals E. coli transcriptome dynamics when exposed to different heavy metal ions. We use a bioinformatics pipeline based on Independent Component Analysis (ICA) to generate insights and hypotheses from this data. We discovered three primary, time-dependent stages of promoter activation to heavy metal stress (fast, intermediate, and steady). Furthermore, we uncovered a global strategy E. coli uses to reallocate resources from stress-related promoters to growth-related promoters following exposure to heavy metal stress.

Over the past few decades, various high-throughput technologies have emerged to investigate how cellular transcription changes in response to environmental perturbations[1,2]. Currently, RNA sequencing (RNA-seq) is the most widely used method to determine the relative abundance of each mRNA transcript in a cellular population[3]. However, RNAseq requires the destruction of cells to harvest RNA, making it challenging to obtain the necessary time resolution to study the dynamics of transcription. Although sampling RNA from parallel cultures at different times can partially overcome this limitation, many protocols only sample mRNA levels at a few time points resulting in a lack of information regarding transcriptional regulatory network (TRN) dynamics[4,5].

To overcome these limitations, we previously introduced Dynomics, a microfluidic platform for monitoring *Escherichia coli* promoter activity in real-time using fluorescence time lapse microscopy[6]. This technology allows us to grow more than 2000 unique bacterial strains continuously in small cell "traps" in the presence of various nutrients and stressors. To monitor the dynamics of the *E. coli* TRN, we combined a pre-existing *E. coli* promoter library with

Dynomics. The promoter library, created by Zaslaver et al., contains approximately 2000 unique *E. coli* strains where each strain has a different, native promoter driving the expression of green fluorescent protein (GFP) on a low-copy number plasmid[7]. This library enables highly accurate dynamic measurements of each promoter in the genome as initially demonstrated in an experiment looking at the response of *E. coli* during a diauxic growth shift[7]. By combining this library with our microfluidic device and a custom imaging platform, we were able to record transcriptional cellular responses to environmental perturbations every 10 minutes for up to 14 days. Using this technology, we observed the dynamics of all *E. coli* promoters with unprecedented temporal resolution.

In the original Dynomics study, Graham et al. explored how *E. coli* responds to heavy metal exposure. Heavy metal contamination, due to industrial wastewater discharge, has become a significant threat to the environment[8]. In response to the presence of these ions, many microorganisms have evolved strategies to cope with high concentrations of toxic heavy metals. For instance, some bacteria express enzymes that can reduce heavy metal ions, lowering their toxicity[9]. The

[1]Department of Bioengineering, University of California San Diego, 9500 Gliman Dr, La Jolla, CA, USA. [2]Novo Nordisk Foundation Center for Biosustainability, Technical University of Denmark, Kemitorvet, Building 220, 2800 Kgs, Lyngby, Denmark. [3]Department of Molecular Biology, School of Biological Sciences, University of California San Diego, 9500 Gliman Dr, La Jolla, CA, USA. [4]Synthetic Biology Institute, University of California San Diego, 9500 Gliman Dr, La Jolla, CA, USA. ✉e-mail: armiano@ucsd.edu

breadth of metal-responsive promoters and genes found throughout the microbial world have made bacteria a popular choice for engineering heavy metal biosensors[6] and bioremediation strategies[10]. While the potential applications of studying bacterial responses to heavy metals have received significant attention, we are interested in the fundamental biological implications of the timing of transcriptional responses to different types of heavy metals. This analysis has the potential to shed light on TRN structure and to help hypothesize on the cellular responses to specific metals based on the temporal patterns of gene activations observed.

Transcriptomic methods such as RNAseq and Dynomics generate complex, high dimensional data sets that require sophisticated analysis techniques to interpret[11]. Independent component analysis (ICA), an unsupervised machine learning technique developed to deconvolute mixed signals into individual sources[12], has proven particularly successful at extracting biologically relevant transcriptional modules across the phylogenetic tree from a variety of transcriptomic data sets[13–17]. In fact, ICA outperformed 42 similar methods in reproducing known transcriptional modules[18]. Using ICA, we can identify groups of genes, called iModulons, which are co-regulated throughout the data and are associated with promoter activity levels in each sample. Generating iModulons has been useful for understanding TRNs and low-resolution dynamics in RNAseq and microarray data[19]. For instance, ICA was able to quantitatively summarize the major steps in *Bacillus subtillis* sporulation by grouping relevant genes into specific iModulons[13]. Moreover, ICA consistently identifies similar gene groupings, even between RNAseq and microarray data sets[19].

In this study, we apply the ICA workflow to the gene expression dataset originally generated by Graham et al., thereby leveraging pre-existing data to derive insights into the temporal patterns of activation of bacterial promoters in response to external heavy metal stress. The data was previously generated using environmentally relevant concentrations of heavy metals, typically several orders of magnitude smaller than the minimum inhibitory concentration (MIC) known for *E. coli*[20]. Our results confirm previously known patterns of promoter activation while also identifying additional gene associations. Importantly, to our knowledge, this is the first time ICA has been applied to a time-series transcriptomic data set with measurements taken at regular short time intervals across multiple days. Combining ICA with high temporal-resolution data enabled us to generate iModulons that reveal the response patterns of different gene groups as a function of time. This approach generates interesting hypotheses regarding transcriptional dynamics.

## Results

### Dynamic transcriptional response to heavy metal exposure

The protocol used to acquire the GFP promoter library data analyzed in this study is described in detail in the original Dynomics publication[6]. In brief, each strain was induced with a heavy metal ion for four hours and then left to recover in minimal media for 20 h (Fig. 1). A custom fluorescence imaging system was used to record the average GFP expression every 10 min. The raw fluorescence data were background-subtracted, median-filtered, and normalized using the mean GFP intensity of a control strain. The fold change was calculated as the log base 2 of the ratio of fluorescence values at selected time points within the induction window to the fluorescence values at the start of induction. To minimize noise while retaining temporal information, six time points spaced forty minutes apart were selected for each induction window. This data was then organized into a matrix, X, where each of 1805 rows represented a different promoter and each of 36 columns represented the log2 of the fold change for each time point during the six heavy metal inductions. Thus, a total of 36 conditions were analyzed for each promoter (6 different heavy metals multiplied by 6 different induction time windows) (Fig. 2a).

ICA was applied to the matrix, X, resulting in an M matrix of promoter coefficients and an A matrix of activity coefficients (Supplementary Data 1–3). The M matrix has dimensions of 1805 by 15, which correspond to 1805 promoters and 15 independently modulated sets of genes referred to as iModulons as suggested in previous studies[14]. The activity profiles and corresponding gene weights for all 15 iModulons obtained from the bioinformatics pipeline are provided in the Supplementary Information (Figs. S1 to 15). Out of the 15 iModulons, six (iModulons 0, 2, 4, 5, 9, and 13) were discarded because they were dominated by a single high coefficient promoter. These iModulons are uninformative and are usually the result of noisy promoters in the data.

While the M matrix shows how much each promoter contributes to each iModulon, the A matrix shows the relative contribution of each heavy metal induction window (e.g., zinc induction and time window two) to each iModulon. We named each iModulon according to its shape and the predominant environmental condition associated with it (Fig. 2b). From the A matrix, we see that each iModulon was largely specific to a single heavy metal suggesting that the underlying biology is highly sensitive to which heavy metal is present (Supplementary Fig. S52).

For lead, iron, cadmium, and chromium, a single iModulon per metal was sufficient to capture the observed response, and these iModulons represent promoters that became steadily activated over the course of the induction. Zinc, on the other hand, had four separate iModulons that captured unique activation patterns. We classified these patterns as "Steady Activation" for promoters whose activity steadily increased over the induction window, "Fast Activation" for promoters that activated quickly after the beginning of the induction, "Intermediate Activation" for promoters that were mostly active in the middle of the activation window, and lastly "Partial Steady Activation" for promoters that were steadily activated at the beginning of the

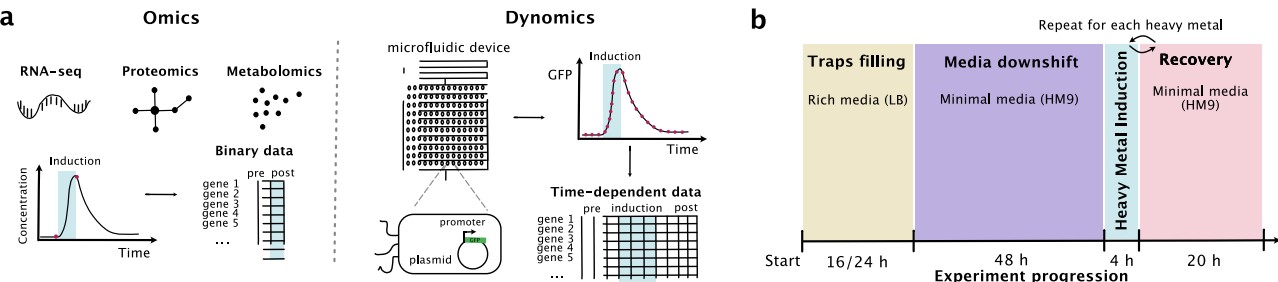

**Fig. 1 | Dynomics experimental set up. a** Illustration of the differences between standard -omics technologies (left) and the Dynomics experimental set up using a 2176-strain microfluidic device (right). **b** Diagram to illustrate the experiment steps. In chronological order the steps are: growth on rich media (LB) until cells completely fill the traps, downshift to minimal media (M9) for 48 h, induction with the heavy metal of choice for 4 h, recovery with minimal media. After 20 h on minimal media another cycle of induction and recovery can be performed.

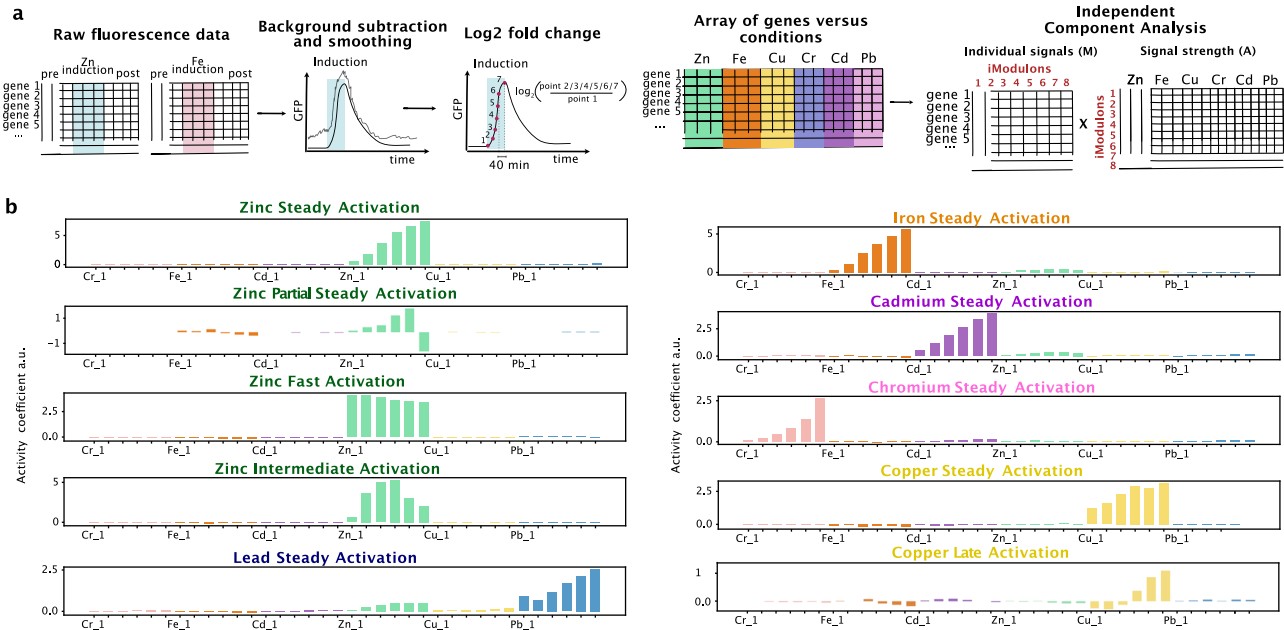

**Fig. 2 | Independent Component Analysis (ICA) to analyze heavy metal inductions data from Dynomics experiments. a** Illustration of the different steps of our data analysis pipeline. The analysis starts with raw fluorescence data which are processed with a background signal removal algorithm, normalized by promoterless strains, and smoothed by median filtering. The log2 is taken to convert to fold change, and the data is formatted as a matrix of genes versus conditions (heavy metal inductions). ICA is applied to this matrix to obtain the M (promoter coefficients) and A (activity coefficients) matrices respectively. **b** iModulon activity plots representing different response dynamics to the heavy metals investigated. The x-axis represents five time points spaced 40 min apart for each of the heavy-metal shown. For illustration purposes only the first time point of each metal is labeled on the x-axis. The y-axis is unitless as it represents the log2 of the fold change of the normalized fluorescence data.

induction window and then suddenly repressed in the last part of the induction window. A summary of the different activation patterns is included in Supplementary Fig. S52. Copper exposure was associated with two iModulons ("Steady" and "Late Activation" iModulons) based on their activation profile pattern. The remaining heavy metal iModulons exhibited a single "Steady Activation" profile.

To identify the most significant promoters in each iModulon, we employed a thresholding method. In this method, we iteratively remove promoters with the highest absolute weighting from an iModulon until the D'Agostino K2 statistic of normality of the remaining distribution falls below a predetermined cutoff. This cutoff distinguishes high-coefficient promoters from those with negligible influence that cluster around zero. By using a fixed threshold value across all iModulons, we minimized bias and ensured a reliable analysis. More details can be found in the original publications that used this thresholding method[21,22]. After applying the calculated cutoffs, we extracted and analyzed the promoters with the highest coefficients for each iModulon. To better understand the overall bacterial response to each heavy metal, we classified the promoters by phenotype (i.e., the functions associated with each promoter based on existing literature) and activation profile (i.e., how and when the promoter is activated) (Fig. 3). In the following sections describing the features of the iModulons, when we state that a certain gene belongs to an iModulon, we mean that the promoter activity for that gene matched the activation profile for that iModulon. We will generally refer to genes as belonging to iModulons for brevity and clarity with the understanding that it is actually the activity of the gene promoters that we are analyzing.

## Zinc iModulons

We observed that the zinc iModulons were associated with several functions, including envelope stress response, oxidative stress response, zinc resistance, the dissimilatory nitrate reduction to ammonium pathway, and protein metabolism (Fig. 3a). Within this iModulon, we identified a set of genes directly related to the envelope

stress response (*yfeY, bacA, ropE*) and to peptidoglycan stress that were present in both fast and steady activation profiles, consistent with previous literature[23–25]. Notably, we also found that two genes associated with steady activation (*mipA, cysQ*) are directly involved in peptidoglycan synthesis. This iModulon also includes the activation of the *glnW* promoter, which transcribes glutamine tRNA. Given that glutamine is important for fatty acid synthesis, we speculate that the upregulation of *glnW* may help restore membrane damage caused by zinc exposure. Our ICA analysis also revealed multiple genes involved in the oxidative stress response, consistent with former studies[26], and distinguished between fast activation (*yedY, yfcG, selC*) and steady activation (*katE, lipA, aldH, nupG*). Interestingly, the promoters for *yedY* and *yfcG* specifically regulate genes that combat oxidative damage[27,28], indicating a targeted response against cellular damage. However, the genes associated with steady activation are involved in cellular processes unrelated to defense response. These include lipoate biosynthesis[29], nucleoside transport[30], and hydrogen peroxide detoxification[31].

Our results also confirm the activation of the *zntA* gene which is associated with zinc export as a detoxification mechanism when excess levels of zinc are detected[32]. Interestingly, our analysis is also able to detect the activation of two genes (*narZ* and *nrfE*) belonging to the dissimilatory nitrate reduction to ammonium pathway (DNRA)[32]. In particular, we were able to observe the temporal dynamics of the pathway activation since *narZ* (nitrate reductase) belonged to the fast activation iModulon and *nrfE* (nitrite reductase) was associated with the intermediate activation profile. The order of activation matches the order required for nitrogen respiration, with nitrate first being reduced to nitrite by *narZ*, then nitrite being reduced to ammonium by *nrfE*[33]. This mechanism follows the "just-in-time" transcription program in metabolic pathways that was previously suggested in the literature[34]. To our knowledge, this is the first time that the activation of DNRA pathway in *E. Coli* is linked to the presence of elevated zinc concentrations. We hypothesize that in this context, DNRA activation

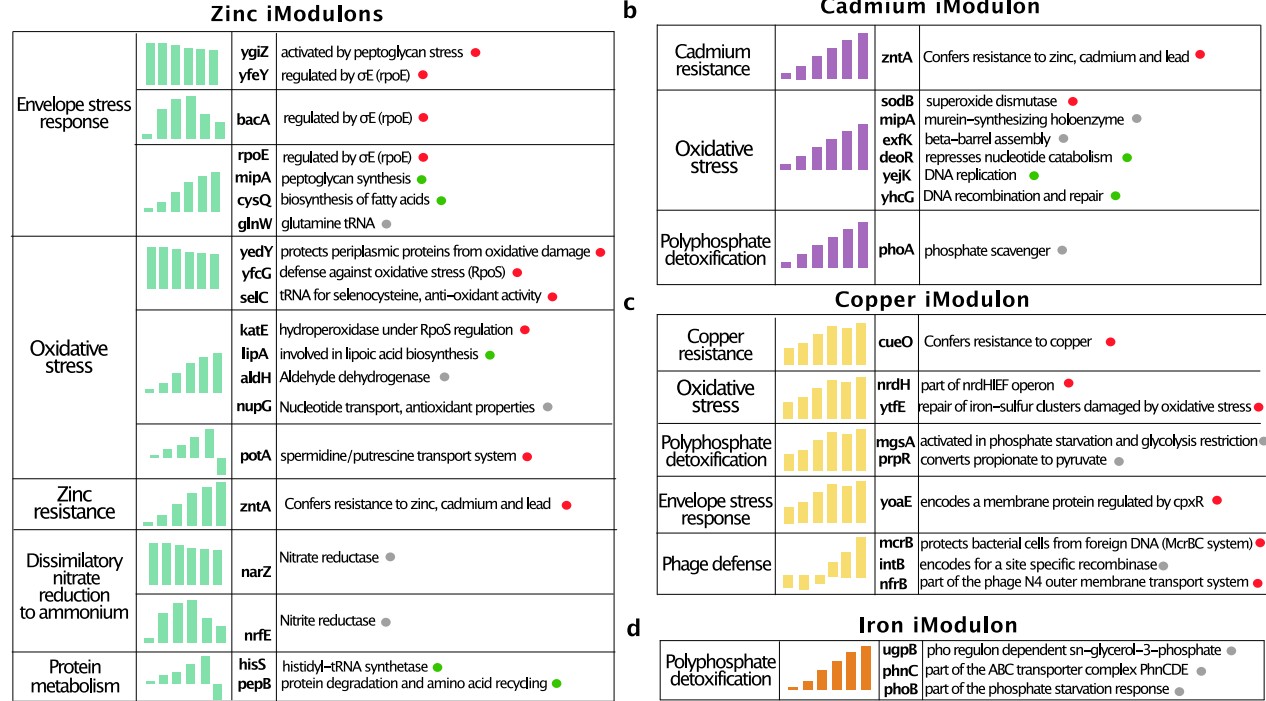

**Fig. 3 | Illustration of significant promoters enriched by the ICA analysis organized by function and activation profile.** Green dots indicate promoters that drive genes related to transcription/translation/synthesis functions. Red dots indicate promoters related to stress/damage responses. Grey dots indicate promoters in neither of those categories. **a** Significant promoters associated to the induction of zinc. Promoters are classified according to the physiological function they are associated with. **b** Significant promoters associated to the induction of cadmium. **c** Significant promoters associated to the induction of copper. **d** Significant promoters associated to the induction of iron.

is beneficial to the cells due to the generation of an electron sink that can be used for NADH re-oxidation into NAD+ which is a key metabolite in counteracting DNA damage and oxidative stress[35,36]. Future experiments will be needed to validate this claim. Finally, we also detect a "Partial Steady Activation" iModulon, characterized by an initial steady response which is followed by a sudden repression in the last phase of the induction window. Interestingly, we find that the promoter which transcribes for the gene *potA* is enriched. This gene is involved in the spermidine and putrescine transport system. Previous studies found that putrescine and spermidine added to the culture medium significantly increased expression of oxidative-stress related genes *oxyR* and *katG*[37]. In line with these findings, we observe that zinc exposure leads to the activation of both *potA* and *katE* promoters (Fig. 3a). The same iModulon enriched for two genes (*hisS*, *pepB*) which are both related to protein metabolism. The *hisS* gene encodes histidyl-tRNA synthetase, an essential enzyme in protein synthesis that charges tRNA with the amino acid histidine[38]. The *pepB* gene encodes endopeptidase PepB or leucyl aminopeptidase, involved in intracellular protein degradation and amino acid recycling[39].

## Cadmium iModulons

The cadmium iModulon was characterized by a steady increase in promoter activation and included promoters that are mainly associated with cadmium resistance and oxidative stress (Fig. 3b). As expected from the literature[32], we found that the *zntA* gene was up-regulated, which confers cadmium tolerance as well as zinc tolerance as previously described. Furthermore, we detected the activation of the *sodB* promoter, which encodes a superoxide dismutase that is a key factor in defending the cell against oxidative stress through decomposition of superoxide radicals. This finding agrees with previous results reported in the literature, which showed that superoxide dismutases (SODs) can protect E. coli from heavy metal toxicity, particularly from cadmium exposure[40].

The iModulon also captures the cell's response to two of the main effects of oxidative stress: lipid peroxidation and DNA damage. In particular, we report the activation of the promoter for *mipA*, which encodes a scaffolding protein for murein synthesizing machinery[41], and the promoter for *exfK* (also known as *bamA*), which is involved in the assembly and insertion of beta-barrel proteins in the outer membrane[42]. Additionally, we identify three promoters which are known to be involved in nucleotide catabolism repression (*deoR*[43]), DNA replication (*yejK*[44]), and DNA recombination and repair (*yhcG*[45]) respectively. Finally, we detect the activation of the *phoA* promoter, which is responsible for the breakdown of organic phosphate esters[46].

## Copper iModulons

The copper iModulon was characterized by a "Steady" activation profile and a "Late" activation profile (Fig. 3c). As expected, the Steady iModulon contains the *cueO* promoter which encodes for a multi-copper oxidase involved in copper tolerance under aerobic conditions[47]. Furthermore, we detect the activation of promoters known to be linked to the oxidative stress response (*nrdH*[48], *ytfE*[49]) and to the envelope stress response (*yoaE*[50]). Interestingly, we also detect the activation of promoter *mgsA* which is known to be activated during phosphate starvation and glycolysis restriction[51]. Similarly, we report the activation of promoter *prpR* which is part of the Pho regulon which we also detected in the iron iModulon[52].

The Late Activation iModulon is characterized by a late response with increased promoter activity starting roughly halfway through the induction window. Interestingly, we find that this iModulon was enriched for multiple genes related to the cell's response against phages including *mcrB*, involved in a restriction-modification system for defense against foreign DNA[53]; *intB*, encoding an integrase enzyme for site-specific recombination[54]; *prpR*, a transcriptional regulator controlling propionate catabolism[55]; and *nfrB*, part of the N4 phage outer membrane transport system involved in phage DNA entry during

infection[56]. It is interesting to note that three of these genes (*mcrB*, *intB*, and *nfrB*) appear to be related to phage defense and interaction, suggesting a potential coordinated response to phage infection or other stressors that could impact the bacterial cell's interaction with foreign DNA. We hypothesize that the activation of genes related to phage defense and interaction might reflect a broader stress response aimed at protecting the cell from additional threats, such as foreign DNA or phage infection, under conditions where cellular processes and genomic integrity are already compromised by copper-induced stress.

## Iron iModulons

The iron iModulon is also characterized by a steady state activation profile (Fig. 3d). Interestingly, we found three enriched genes (*ugpB*, *phnC* and *phoB*) which all belong to the Pho regulon, a group of genes in *E. coli* that are involved in the regulation of phosphate metabolism[57]. In particular, *phoB* is the response regulator in a two component regulatory system with *phoR* (or *creC*) and regulates inorganic phosphate ($P_i$) uptake. The gene *phnC* is part of the ABC transporter complex PhnCDE involved in phosphonates, phosphate esters, phosphite and phosphate import. Finally, gene *ugpB* codes for a binding protein-dependent sn-glycerol-3-phosphate transport system which is under the control of the Pho regulon. In particular, ugp-dependent G3P transport activity is present only after growth at limiting concentrations of $P_i$[58]. Previous literature indicates that excess iron can cause the formation of toxic reactive oxygen species (ROS) through Fenton chemistry[59]. Both iron stress and oxidative DNA damage are successfully prevented by polyphosphates which are polymers formed by covalently linked inorganic phosphates[59]. Therefore, we hypothesize that the genes enriched in the Iron iModulon reflect the need for the cell to import additional phosphate in order to form polyphosphates which can be used as a defense mechanism against iron. When excess iron is present, *E. coli* can transport it into the cell and store it within the polyphosphate granules, which act as a sink for the iron[60]. Based on the lack of specific iron toxicity genes in the iModulon, we propose that the concentration of iron used was too low to induce specific responses. Nonetheless, it is still informative that phosphate chemistry accounts for the ROS-related effects of an increase in iron.

## Transcriptional response of the recovery post-induction

The primary advantage of the Dynomics technology is the ability to track promoter activity over an extended period of time. This enabled us to analyze not only the effect of the heavy metal exposure during the induction window, but also to explore how the bacteria responded during the recovery period post-induction. In this case, we calculated the log2 of the fold change with respect to the end of the induction window (Fig. 4a). We took into consideration 20 time points spaced 40 minutes apart (Supplementary Data 4). Applying the ICA algorithm to this dataset we found 35 iModulons (Figs. 4b, S16 to 50, Supplementary Data 5 and 6). We focus our analysis on a subset of iModulons whose activity seemed particularly interesting and biologically significant (Fig. 4c, Methods).

## Zinc iModulons

We report four iModulons whose activity profiles show activation following zinc exposure. In line with the induction response, we identify a few promoters associated with stress responses such as the SOS response (*recX*), peptoglycan stress (*yjfQ*, *helD*, *skp*) and oxidative stress (*katE*). We also report the activation of promoters which are related to transcription and translation processes such us RNA processing and decay (*rne*), purine metabolism (*yneF*), proline tRNAs (*proL*), methionine synthesis (*metF*), and DNA ligation (*ligB*). Additionally, we find that one of the zinc-related iModulons (iModulon 16) transcribes mRNA for monothiol glutaredoxin (*ydhD*, also known as *grxD*) which is involved in the biogenesis of iron-sulfur clusters[61]. This

confirms previous studies that have shown that excess zinc is associated with the disruption of iron–sulfur clusters in *E. coli*[62]. Interestingly, the same iModulon contains the *edd* promoter which encodes the enzyme phosphogluconate dehydratase which is the first reaction in the Entner-Doudoroff pathway and uses the [4Fe–4S] iron-sulfur cluster as a cofactor[63].

## Cadmium iModulons

The ICA analysis also identifies two iModulons whose activity profiles are mostly active post cadmium exposure (Fig. 4c). Similar to the trend we observed for the zinc iModulons, we find that there are several promoters associated with biosynthesis and protein translation processes. This bolsters our prediction that the cells mainly activate stress response processes during the induction itself and switch to recovery and growth processes when the inducer is removed. In particular, for the post-induction cadmium iModulon we report the activation of promoter *asd* which is involved in L-lysine, L-methionine, and L-threonine biosynthesis, promoter *tufB* which codes for an elongation factor for protein biosynthesis and *ykgM* which encodes a ribosomal protein. Interestingly, *ykgM* was previously found to be upregulated upon zinc starvation[64,65]. This physiological response is further supported by the activation of promoter *znuC* which drives expression of a zinc import ATP-binding protein. Our results therefore align with previous studies based on the analysis of genome-wide temporal gene expression data which suggested that the molecular mechanisms of cadmium toxicity could be partially explained by the disruption in the transcription of genes encoding ribosomal proteins and zinc-binding proteins[66].

## Iron iModulons

Furthermore, we report two iModulons containing promoters that activate following iron induction. We find that one of the promoters enriched in these iModulons is *ompW* which transcribes mRNA for an outer membrane protein whose expression has been previously observed to be down-regulated in iron limited growth conditions[67]. Our results show that the expression of *ompW* is greatly increased post-induction when the toxin is removed from the media. Additionally, we find a second iron iModulon with a steeper activation profile which is associated with the activation of two genes: *rfbB* (involved in biogenesis of the bacterial outer membrane) and *ydcL* (uncharacterized lipoprotein).

## Copper iModulons

Finally, we report a copper recovery iModulon linked to the activation of five genes, two of which are involved in amino acid biosynthesis (*metF*) and DNA replication/transcription processes (*helD*). We categorized the promoters according to their function in three main groups which are identified with dots of different colors in Fig. 4.

Analyzing the data across the induction and post-induction window, we report a clear shift from an enrichment of promoters driving genes related to stress and damage responses (45% during induction versus 23% post induction) to promoters associated with transcription, translation and synthesis functions (18% during induction versus 42% post induction)(Fig. 4d).

# Discussion

This study presents the integration of a genome-scale platform for monitoring temporal gene expression with the analytical power of ICA[6,14]. Previous studies showed that ICA is highly effective at extracting biologically significant transcriptional modules (called iModulons) from a wide range of datasets, but these datasets lacked information on the time-dependent activation of genes. Instead, most studies have relied on static data obtained from RNA-seq, proteomics, and metabolomics experiments[13,19,68,69]. The Dynomics microfluidic platform enabled sampling of promoter activity every 10 min for

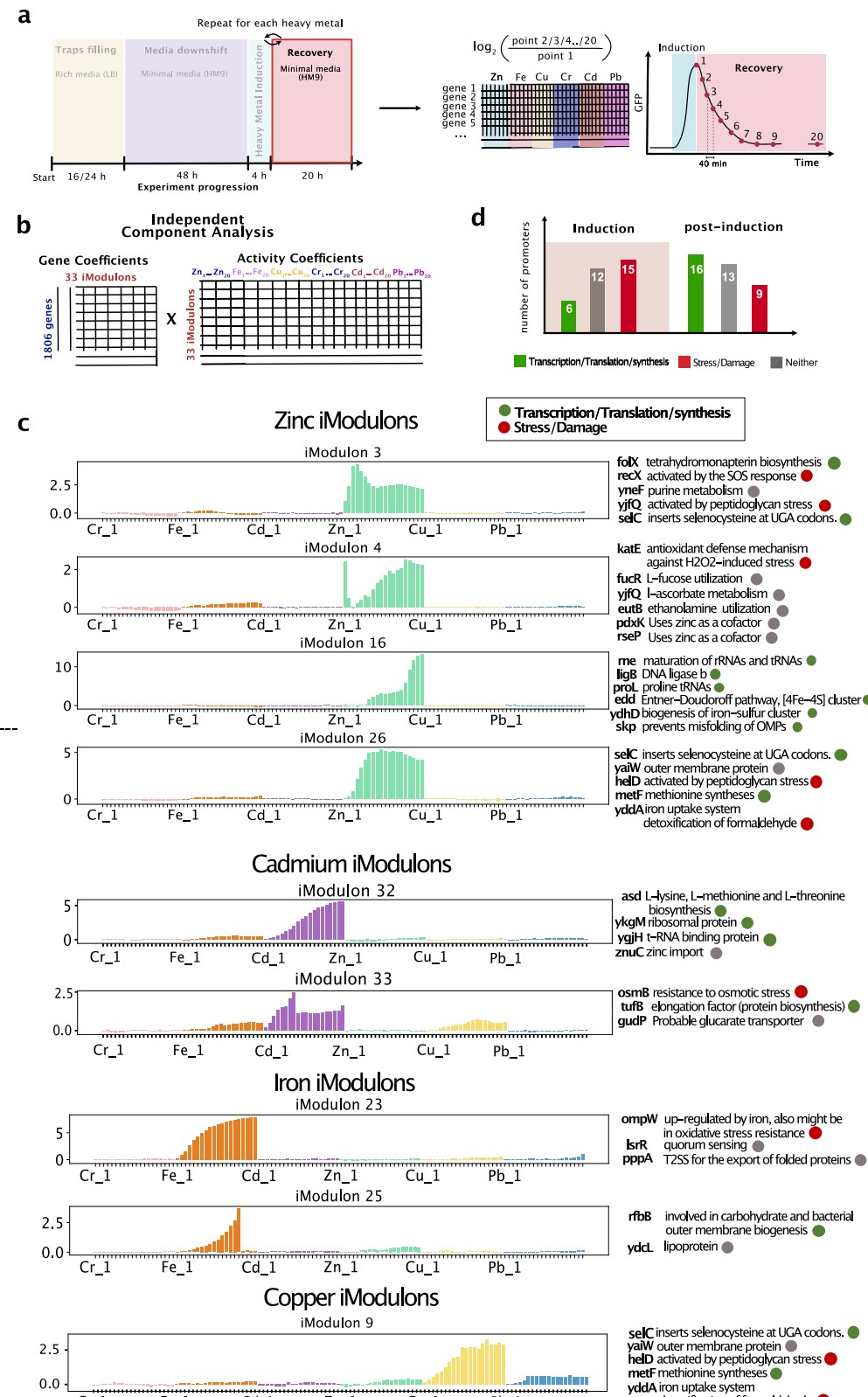

**Fig. 4 | ICA analysis of the recovery period following heavy metal inductions.**
**a** Diagram illustrating the steps involved in the analysis pipeline. The ICA was applied to the data collected from the end of the induction window. We considered 20 time points at a distance of 40 min each. **b** Matrix decomposition after applying ICA. Overall, the ICA analysis identifies 33 iModulons, a selection of which is shown in panel C. **c** Activity plots and related significant genes for a selection of iModulons associated to the post-induction response. The x-axis represents five time points spaced 40 min apart for each of the heavy-metal shown. For illustration purposes only the first time point of each metal is labeled. The y-axis is unitless as it represents the log 2 of the fold change of the normalized fluorescence data. Green dots indicate promoters that drive genes related to transcription/translation/synthesis functions. Red dots indicate promoters related to stress/damage responses. Grey dots indicate promoters in neither of those categories. **d** Comparison of number of promoters grouped by function during and after heavy metal induction.

multiple days, which is unrivaled by any transcriptome-scale analysis method to our knowledge.

By applying ICA to this data set, we demonstrated the importance of time-dependent analysis in providing insights into the dynamic nature of gene expression in response to environmental stressors. By splitting different heavy metal inductions into separate 40 minute time windows, we were able to apply ICA to time-series transcriptomic data for the first time. We observed the richest response for the zinc inductions. Specifically, we found four different iModulons which differentiate fast responders (genes that are activated at the start of the induction window) from intermediate responders (genes that are maximally active in the middle of the induction window), steady responders (genes whose expression steadily increases throughout the induction window) and partially steady responders (genes that steadily increase over time until they are repressed in the last window). The zinc data demonstrates the ability of this analysis method to resolve the activation sequence of promoters involved in the same metabolic pathway. We detect the activation of promoter *narZ* as an early responder and promoter *nrfE* as a late responder which are involved in the first and second step of dissimilatory nitrate reduction to ammonium metabolic pathway. This result is a clear example of the power of this platform when used for metabolic pathway reconstruction which is a topic of great interest in systems biology[70].

Additionally, the data in this study were generated using environmentally relevant heavy metal concentrations which are significantly lower than the concentration range known to be toxic for *E. coli*[71]. This allowed us to study the bacterial response to elevated, but non-toxic, levels of iron in the environment. Our ICA analysis suggests that in this scenario the bacteria activate several genes in the Pho regulon (which regulates phosphate uptake and metabolism) in response to elevated concentrations of iron. We hypothesize that bacteria accumulate phosphate as an early response to excess iron in preparation for polyphosphate synthesis. Several studies have shown that polyphosphates can sequester iron in *E. coli* and other microorganisms, limiting its bioavailability and protecting cells from iron toxicity[59,72]. Further studies need to be conducted to experimentally verify this hypothesis.

Lastly, this study expands our understanding of the recovery process *E. coli* following the removal of stressors such as heavy metals. Our findings indicate a marked shift in the cellular functions of enriched promoters during and after heavy metal induction. We quantified a transition from the activation of promoters associated with stress defense mechanisms and detoxification processes to the activation of promoters involved in ribosome biosynthesis, tRNA synthesis, mRNA processing and decay, amino acid biosynthesis, and replication.

While the library of 1805 *E. coli* promoters employed in this study represents the most comprehensive collection currently available and allows for substantial insights into gene expression dynamics, we acknowledge that it encompasses less than half of the total known promoters in *E. coli*. This limitation signifies that our analysis might not fully capture all the potential regulatory mechanisms at play, and we advocate for future studies to incorporate a more expansive promoter library to offer a more encompassing view of *E. coli* gene expression.

Using concentrations that are several orders of magnitude lower than the minimum inhibitory concentrations (MICs) for *E. coli* led us to the observe distinct gene expression patterns compared to those reported in previous studies[66,71,73,74]. We speculate that by utilizing lower heavy metal concentrations, we are better able to reveal the subtle and nuanced responses of bacteria to these stressors, which might otherwise be overshadowed by the more pronounced effects at higher concentrations. This, in turn, contributes to a more comprehensive understanding of bacterial behavior and adaptation mechanisms under more environmentally relevant conditions, expanding our knowledge of how bacteria cope with heavy metal exposure in real-world scenarios. Overall, we believe that combination of Dynomics

promoter activity data with our approach to applying ICA to time-series data will continue to serve as a valuable tool for generating hypotheses on how cells respond to various external stimuli, uncovering fundamental principles of transcriptional regulation.

## Methods

### Data collection
Detailed description of the Dynomics experimental set up can be found in the original study[6]. In brief, data was obtained from fluorescence values extraction from flat-field corrected images gathered using a custom optical set up. The first step of the experimental set up is arraying the cells using a Singer ROTOR robot so that they could be spotted onto the microfluidic device before glass bonding. Once the device was ready, it was set up inside a custom box kept at 37 °C for imaging. For media flow, the inlet and outlet were connected to 140 mL syringes. The concentrations of the heavy metals tested in this study can be found in Fig. S51.

### Data processing
Detailed information on how the raw data was processed can be found in the original study[6]. In summary, the data was first processed by subtracting the local background signal and then dividing the result by the background signal again in order to create a measure of the amplification of the signal over the background. Specifically, fluorescence values measured at a location outside the cell trap were subtracted from those obtained within the cell-containing regions. This method effectively removes signal noise not associated with the cells, thereby providing a more reliable representation of the cellular fluorescence signals. Then, the data were passed through a median filter (scipy.signal.medfilt, kernel_size = 11) and normalized by subtracting and dividing the average expression values of the promoterless strains. The code used to process data from the original study is available on GitHub. The original data files post-processing for each heavy metal are also available on GitHub. The data were further processed by calculating the log base 2 of the fold change of the ratio of six time points (spaced 40 min apart) with respect to the start of the induction window for the data plotted in Figs. 2 and 3. On the other hand, for the dataset used to produce the results shown in Fig. 4 the data was converted in the log base 2 of the fold change of the ratio between 20 points (spaced 40 minutes apart) with respect to the end of the induction window. All python scripts used to produce these datasets are available in the "ICA_dynomics" GitHub repository. The final dataset which was fed to the ICA algorithm consisted of a matrix where the rows represented all the different promoters (1805) and the columns represented the different conditions (i.e. the type of metal and the specific time point). Therefore, for the analysis in Figs. 1 and 2 the final dataset had a total of 36 columns representing 6 heavy metal inductions, each with 6 time points. Similarly, the dataset behind the results of Fig. 4 had dimensions of 1805 rows and 100 columns representing 6 heavy metal inductions, each with 20 points.

### Independent component analysis
We used the pipeline for ICA implementation that has been described in previous studies[14,22]. In brief, we run the *Scikit-learn70* (v0.19.2) implementation of the FastICA algorithm 100 times with random seeds, a convergence tolerance of $10^{-6}$. The number of components in each iteration was set to the number of components that reconstruct 99% of the variance as calculated by principal component analysis. The resulting source components (M) from all runs were clustered using the Scikit-learn implementation of the DBSCAN algorithm which does not require predetermination of the number of clusters. In our DBSCAN analysis, we used the following parameters: DISTANCE parameter, analogous to the epsilon parameter in traditional DBSCAN applications, was set to its default value of 0.1, determining the maximum distance between two points to be considered as in the same

cluster. Meanwhile, the MIN_FRAC parameter, which specifies the minimum fraction of total data points required to form a valid cluster, was maintained at its default setting of 0.5. This implies that each cluster in our analysis contains at least 50% of the total data points present in the dataset. The final independent components were defined as the centroid of each cluster in M, and the weightings were defined as the centroid of their corresponding weighting vectors in A. To ensure that the final components were consistent across multiple runs, we computed the clustered components 100 times, and selected the components that were identified in every run. The previously published code used to compute robust independent components is publicly available at github.com/SBRG/precise-db. We also added the scripts specifically used in this paper in the "precise_db" folder within the "ICA_dynomics" GitHub repository.

### Determination of the gene coefficient threshold

The dataset M contains all the genes coefficients associated with each iModulon. Most of these coefficients have values close to zero which indicates they are not significantly enriched for that iModulon. In order to extract the genes that belong to each iModulon we computed the D'Agostino $K^2$ test statistic which is a measure of the skew and kurtosis of a sample distribution. Genes with the largest absolute value were iteratively removed and the D'Agostino K2 test statistic was computed for the resulting distribution[14]. The statistic cutoff was kept fixed at 800 for the entire analysis. We only considered positively correlated genes in this analysis. The code to calculate the threshold was originally developed by the Palsson lab at UCSD and can be found at the GitHub repository pymodulon. We also included the scripts used specifically for this paper in the pymodulon folder at the ICA_dynomics repository.

### Identification of primary heavy metal associated to each iModulon

We used a simple computational strategy that involved the analysis of the activation matrix data of each iModulon to associate them with the heavy metal that induced the highest aggregate expression over the six time points within the corresponding induction window. This approach allowed us to pinpoint the condition under which each iModulon exhibited the most pronounced change, thereby facilitating a data-driven classification. The script used to implement this classification can be found in the ICA_dynomics GitHub repository.

### Classification of iModulons based on patterns in activity coefficients

In our study, we employed a simple computational method to classify the iModulons based on the shape of their activity coefficients within the induction window of heavy metal enrichment. This method involved calculating the center of mass (CM) and identifying the highest peak (HP) within the specified window for each iModulon, wherein the center of mass represents the mean position of a given function. The iModulons were categorized into three groups based on these calculations: Steady (CM ≤ 5 and HP ≥ 4), Intermediate (CM ≤ 4 and HP ≤ 4), and Fast (CM ≤ 4 and HP ≤ 3). iModulons that did not conform to these parameters were labeled as "other," with plans for further characterization on a case-by-case basis. We have made the corresponding code block available in the "iModulon_identification" script within our GitHub repository, and illustrated the calculated values and resultant classifications in Supplementary Fig. S54. It's noteworthy that the methodology applied is tailored to the specific shapes encountered in our study. For future studies, especially where the shape of activity coefficients is unknown or difficult to categorize, we advocate for exploring more generalized approaches. Machine learning methodologies, particularly those centered around shape recognition, could offer a robust and adaptable solution for classifying transcriptional profiles regardless of the inherent patterns within the data.

### Reporting summary

Further information on research design is available in the Nature Portfolio Reporting Summary linked to this article.

## Data availability

All data is available in the paper, Supplementary Materials and on Figshare.

## Code availability

The code used is available on GitHub with identifier *ICA_dynomics*. Citation: Miano, A. (2023). High-Resolution Temporal Profiling of E. coli Transcriptional Response. Zenodo. We also link the other repositories which were used for different parts of data processing: •`precise-db` - https://github.com/SBRG/precise-db •`pymodulon`—https://github.com/SBRG/pymodulon •`dynomics_public`—https://github.com/GarrettCGraham/dynomics_public.

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

## Acknowledgements

We would like to acknowledge Garrett Graham and Nicholas Csicsery for their help and mentorship in the initial stages of this project. Additionally, we would like to thank Elizabeth Stasiowski and Gregoire Thouvenin for kindly sharing the data used in this project and Professor Terence Hwa for his helpful comments and advice on the project. This work was supported by the National Institute of General Medical Sciences of the National Institutes of Health grant No. RO1GM069811 (J.H.).

## Author contributions

Conceptualization: A.M. and J.H. Methodology: A.M., K.R., A.S., B.P., and J.H. Analysis: A.M., K.R., and A.S. Writing—original draft: A.M. Writing—review & editing: A.M., A.L., and K.R.

## Competing interests

J.H. is a co-founder of GenCirq Inc., which focus on cancer therapeutics. He is on the Board of Directors and has equity in GenCirq. His spouse is employed part time for bookkeeping and to support employees with Human Resources.The remaining authors declare no competing interests.
