## [Peer Review File NEW · Nature Communications]

High-Resolution Temporal Profiling of E. coli Transcriptional ResponseEditorial Note: Parts of this Peer Review File have been redacted as indicated to remove third-party material where no permission to publish could be obtained.

Reviewer #1 (Remarks to the Author):

The paper titled High-Resolution Temporal Profiling of E. coli Transcriptional Response examined the temporal dynamics of the E. coli transcriptome when exposed to different heavy metal ions. Using ICA, the authors found distinct iModulons (set of promoters) that capture the transcriptional response to a specific heavy metal. They also discovered promoters had one of three activation dynamic responses to heavy metal stress.

When coupling the previous Dynamics technology, which is able to capture temporal transcriptional patterns, with the author's ICA approach researchers can make hypotheses about the biology driving the dynamics and learn. For example, since the zinc iModulon includes glnW promoter, which transcribes glutamine tRNA and is important for fatty acid synthesis, they hypothesized that the upregulation of glnW may help restore membrane damage caused by zinc exposure. Having this approach as a hypothesis generation tool is significant and helpful for furthering our understanding about the transcriptional response of bacteria to stimuli/other environmental stressors, which can ultimately help to improve treatment development or synthetic engineering. Overall this paper was very enjoyable to read and it was exciting to see the use of ICA on temporal transcriptional data.

Minor changes

1. In terms of the methodology, I'm curious if the authors tried other dimensionality reduction/representation learning methods such as NMF using their data?

2. While they provided their github repository (<https://github.com/SBRG/precise-db>), which allows users to perform ICA on their own gene expression data to identify modules, there didn't appear to be scripts that were used to reproduce the results presented in the paper. The description in the methods does provide most of the detail necessary, but there are some details that remained unclear including:

2a. What is meant by correcting for "local background signal"?

2b. What are the parameter settings for clustering is unknown

2c. How robust were the ICA results across runs? Were there the same number of components per run? Were the components the same per run?

2d. How did they determine which iModulon correspond to which condition based on matrix A?

2e. How were the transcriptional profiles (i.e steady, fast) determined?

2f. Please elaborate on the reasoning for the filtering (the data were passed through a median filter (`scipy.signal.medfilt`, kernel size = 11)) and then sub sampled to time points 40 minutes apart. Why not use all the time points? Why the overlapping filter windows? Were these preprocessing hyper parameters varied?

3. They claim that their approach will reveal response patterns of different gene groups as a function of time, which they did. They also claim that their ICA method can reveal details about TRN structure, cellular stresses associated with specific metals, and important relationships between co-expressed genes. However, there were no results demonstrating this claim about using this approach to identify gene-gene relationships.

Reviewer #2 (Remarks to the Author):

The work from Miano et al. describes the analysis of temporal gene expression data obtained in response to heavy metal ions. Temporal high-throughput data is obtained by using the Dynamics platform, as previously reported [PMID: 31974311] by one of the authors (Jeff Hasty) of this manuscript. Temporal data is analyzed using the iModulons framework to generate insights into the temporal activation patterns of bacterial promoters in response to external heavy metal stress.

Major concerns

- From the manuscript, it is not clear whether the authors applied the ICA workflow to the gene expression data previously obtained by Graham et al. Or whether they redid the experiments to obtain new data. This point must be clarified in the manuscript.

- Lines 14-20: Dynamics was used with a pre-existing library of 2000 E. coli promoters. It is the most complete promoters' library available, but 2000 promoters are less than the half of total promoters in E. coli. The authors must assess how this incompleteness affects conclusions.
- The methodology is hard to follow for a reader not familiar with the previous papers by the authors. There are gaps and imprecisions that the authors must address. This is key for providing a transparent methodology, thus enabling reproducibility.

Minor concerns

- Line 73: Figure 1 is missing.
- Lines 104-109: The authors classified the different types of activation patterns into 4 classes. A table or figure supporting these definitions will improve clarity.
- Lines 246-304: The results discussed in the section "Transcriptional response of the recovery post-induction" should use the same sub-headers structure (by iModulon) that the "Dynamic transcriptional response to heavy metal exposure" section to improve reading.

General Comments to the Referees

We appreciate the time and effort expended by the reviewers in scrutinizing our manuscript. Their insightful comments and suggestions have immensely helped in enhancing the completeness and precision of this paper. We believe that the manuscript has significantly improved owing to their contributions.

Response to Reviewer 1

1. In terms of the methodology, I'm curious if the authors tried other dimensionality reduction/representation learning methods such as NMF using their data?

We thank the reviewer for this question about exploring other dimensionality reduction and representation learning methods such as NMF. However, we opted to directly utilize Independent Component Analysis (ICA) from the outset due to its well-documented efficacy in handling datasets similar to ours.

In the realm of transcriptional dynamics, Independent Component Analysis (ICA) offers several compelling advantages over Non-negative Matrix Factorization (NMF) that make it a particularly attractive choice for disentangling the complex relationships within gene expression data. One of the most significant strengths of ICA lies in its ability to isolate statistically independent components, which can reveal distinct transcriptional programs or pathways that are activated across a range of biological samples. This capability is crucial when investigating dynamically changing processes such as cellular differentiation or response to stimuli. Unlike NMF, ICA allows for both positive and negative values in its components, enabling the method to capture both upregulation and downregulation of genes. In this paper we found that this dual-sided representation was indispensable for understanding the full spectrum of gene expression changes as many of the iModulons are characterized by activation profiles that switch from activation to repression within the induction window. (Stein-O'Brien, Genevieve L., et al. "Enter the matrix: factorization uncovers knowledge from omics." *Trends in Genetics* 34.10 (2018): 790-805.)

In substantiating our choice, we also relied on recent extensive reviews in the field which analyzed 42 different module detection methods and conclusively demonstrated that ICA surpassed other algorithms in pinpointing groups of co-regulated genes. This research has provided a strong endorsement for the use of ICA in identifying coregulated genes effectively, highlighting its superiority in discerning underlying patterns and structures in complex biological data (Saelens, W., Cannoodt, R. & Saeys, Y. A comprehensive evaluation of module detection methods for gene expression data. *Nat. Commun.* 9, 1090 (2018).)

We are confident that leveraging ICA in our study allows for a robust and optimal analysis, facilitating the accurate identification of co-regulated genes from our temporal dataset.

2. While they provided their github repository (<https://github.com/SBRG/precise-db>), which allows users to perform ICA on their own gene expression data to identify modules, there didn't appear to be scripts that were used to reproduce the results presented in the paper. The description in the methods does provide most of the detail necessary, but there are some details that remained unclear including:

Thank you for bringing this to our attention. We are committed to maintaining a high standard of transparency and reproducibility in our research.

In response to your feedback, we have created our GitHub repository (ica_dynamics - https://github.com/armiano/ICA_dynamics) to include all the scripts utilized to produce the results presented in the paper. We believe that this will not only clarify the specific details you found unclear but will also facilitate a deeper understanding of the methodologies employed in our research for all readers.

2a. What is meant by correcting for "local background signal"?

We thank the reviewer for this question regarding the correction for "local background signal" in our research.

In line with the approach utilized in the original study (Graham et al, 2020) , which we have referenced in our manuscript, the data was corrected for the "local background signal". This term refers to the unwanted noise and signals that are not emanating from the cells (bulb region in the microfluidic device) but from other sources such as the autofluorescence from the microfluidics device and regional fluctuations in the signal due to optics.

The fluorescence values at locations outside of the cell-containing regions was measured, which exhibit the background signals. Then this value is subtracted from the fluorescence values recorded within the bulb (cell-containing traps), effectively eliminating noise that is not associated with the cells.

Figure S3 in the original study (Graham et al, 2020) clearly shows where the background signal was taken from.

Redacted

We added additional text in the Methods section so clarify this point as follows:

“Specifically, fluorescence values measured at a location outside the cell trap were subtracted from those obtained within the cell-containing regions. This method effectively removes signal noise not associated with the cells, thereby providing a more reliable representation of the cellular fluorescence signals.”

2b. What are the parameter settings for clustering is unknown

We thank the reviewer for this question regarding the parameter settings used in our clustering analysis.

To facilitate a comprehensive understanding and transparency of our process, we have detailed all the parameter settings utilized in the methods section of our manuscript as follows:

Line 360: “In our DBSCAN analysis, we used the following parameters: the DISTANCE parameter, analogous to the epsilon (ϵ) parameter in traditional DBSCAN applications, was set to its default value of 0.1, determining the maximum distance between two points to be considered as in the same cluster. Meanwhile, the MIN_FRAC parameter, which specifies the minimum fraction of total data points required to form a valid cluster, was maintained at its default setting of 0.5. This implies that each cluster in our analysis contains at least 50% of the total data points present in the dataset.”

Furthermore, all scripts (including the parameters used) can be found in the GitHub repository.

2c. How robust were the ICA results across runs? Were there the same number of components per run? Were the components the same per run?

We thank the reviewer for this question concerning the robustness of the ICA results across runs.

In our study, the "final independent components" we kept were not arbitrarily selected but were carefully identified as the centroids of each data cluster in matrix M (Sastry et al. 2019). Similarly, the weightings for these components were determined by locating the centroids of their respective weighting vectors in another matrix, labeled A. This systematic approach enabled us to pinpoint key variables with higher confidence.

To rigorously assess the robustness and reliability of these final independent components, we conducted multiple iterations of the clustering process—specifically, 100 separate runs. The objective was to verify that these components were not artifacts of a single computational run but were instead consistently appearing patterns in the data. We found that the same components emerged in each of the 100 runs, substantiating their robustness and reaffirming their importance. The components that were consistently identified across all runs were selected as the final robust components for further analysis. Therefore, to directly answer the questions posed: yes, there was a consistent number of components identified per run, and these components were the same across all runs, reinforcing the robustness and reliability of our findings.

We added the following text to the Methods section to clarify how the final components were determined:

Line 367: "The final independent components were defined as the centroid of each cluster in M, and the weightings were defined as the centroid of their corresponding weighting vectors in A. To ensure that the final components were consistent across multiple runs, we computed the clustered components 100 times, and selected the components that were identified in every run."

2d. How did they determine which iModulon correspond to which condition based on matrix A?

We appreciate your question on the specific methodology utilized to correlate iModulons with respective conditions based on the information derived from matrix A.

We used a simple computational strategy that involved the analysis of the activation matrix data of each iModulon to associate them with the heavy metal that induced the highest aggregate expression over the six time points within the corresponding induction window. This approach allowed us to pinpoint the condition under which each iModulon exhibited the most pronounced change, thereby facilitating a data-driven classification.

In the manuscript, we predominantly showcase and analyze iModulons associated with a single primary heavy metal, identified based on the highest amplitude in the induction windows.

However, it is worth noting that a minority of iModulons exhibit activity across multiple induction windows corresponding to different heavy metals.

While our primary focus remains on the iModulons linked to a main heavy metal, our identification script is equipped to detect these secondary heavy metals as well. This additional data is available in the supplementary materials for reference. We have chosen not to delve deeply into the analysis of these secondary associations in the main manuscript, as they introduce a layer of complexity in biological interpretation. Secondary associations, though present, were not as strong and would require further experimental verification to be substantiated which is outside the scope of this project. Nonetheless, this information is valuable for a more comprehensive understanding and can be a foundation for future detailed studies exploring the intricate dynamics of microbial responses to various heavy metals.

To ensure transparency and reproducibility, we included a detailed figure delineating this classification in the supplementary material (Figure S53). Moreover, we added the Python script (“iModulon_identification”) utilized for this analysis in our GitHub repository.

We also added an additional section in the Methods section as follows:

Line 388: “ Identification of primary heavy metal associated to each iModulon

We used a simple computational strategy that involved the analysis of the activation matrix data of each iModulon to associate them with the heavy metal that induced the highest aggregate expression over the six time points within the corresponding induction window. This approach allowed us to pinpoint the condition under which each iModulon exhibited the most pronounced

change, thereby facilitating a data-driven classification. The script used to implement this classification can be found in the ICA_dynamics GitHub repository. “

2e. How were the transcriptional profiles (i.e steady, fast) determined?

Thank you for your inquiry regarding the methodology used for determining transcriptional profiles in our study. We devised a straightforward computational method to categorize iModulons based on the shape of their activity coefficients within the induction window of heavy metal enrichment. This method involved calculating the center of mass (CM) and identifying the highest peak (HP) within the specified window for each iModulon, with the center of mass representing the mean position of a given function.

Based on these calculations, we set rules to classify the iModulons into three categories:

- **Steady:** Characterized by a $CM \leq 5$ and a $HP \geq 4$.
- **Intermediate:** Defined by a $CM \leq 4$ and a $HP \leq 4$.
- **Fast:** Identified by a $CM \leq 4$ and a $HP \leq 3$.

iModulons not conforming to these parameters were categorized as "other," with plans for further characterization on a case-by-case basis.

We have made the code block available in the "iModulon_identification" script within our GitHub repository for reference. Additionally, the calculated values and the resultant classifications are illustrated in Supplementary Figure S54 (below).

iModulon	Enriched_Metal	center_of_mass	highest_peak	shape_classification
0	Pb	6.00937124004475	6	Mixed Activation
1	Zn	1.9586905603404643	5	Steady Activation
2	Fe	2.8858609181047212	4	Intermediate Activation
3	Cd	3.400290595224472	6	Steady Activation
4	Cr	2.797106658835455	5	Steady Activation
5	Pb	3.226218187718445	6	Steady Activation
6	Fe	3.577980624456813	6	Steady Activation
7	Zn	3.485144831724121	6	Steady Activation
8	Zn	2.382074032444398	2	Fast Activation
9	Zn	3.0220339595067416	6	Steady Activation
10	Cu	2.994608670476212	6	Steady Activation
11	Cr	4.017869426039383	6	Steady Activation
12	Zn	2.6424086644095337	4	Intermediate Activation
13	Fe	1.9954508079486497	2	Fast Activation
14	Cu	5.714600247227268	6	Mixed Activation

We acknowledge that the method employed is tailored to the specific shapes encountered in our study. For future studies, particularly where the shape of the activity coefficients is unknown or challenging to categorize, we recommend exploring more generalized approaches. Machine

learning methods, particularly those focusing on shape recognition, could provide a robust and adaptable solution for classifying transcriptional profiles irrespective of the data's inherent patterns.

We appreciate the opportunity to clarify our methodology and remain open to providing further information or engaging in additional discussions on this matter.

2f. Please elaborate on the reasoning for the filtering (the data were passed through a median filter (`scipy.signal.medfilt`, kernel size = 11)) and then sub sampled to time points 40 minutes apart. Why not use all the time points? Why the overlapping filter windows? Were these preprocessing hyper parameters varied?

We thank the reviewer for this question about data processing.

Overall, these preprocessing parameters were kept unchanged for all analysis presented in the paper.

By leveraging the median filter, we are targeting and eliminating the sporadic fluctuations that occur at singular time points, which are more often than not, a result of equipment irregularities such as flow inconsistencies and optical disturbances, rather than being indicative of genuine transcriptional alterations which in our data characteristically manifest over several time points (Graham et al.). By smoothing the data through median filtering, we create a more stable dataset, devoid of transient noise, facilitating a more reliable application of log₂ fold change analysis without the risk of inadvertently amplifying noise.

When applying ICA, we aimed at determining the optimal effective data dimension which would balance between under and over decomposition. Indeed, choosing the effective dimension too high might lead to signal-to-noise ratio deterioration, overfitting and splitting of the meaningful components. This is usually noticed by the presence of iModulons dominated by a single, high-coefficient gene. Based on this, we constructed the dataset that lead to biologically meaningful and interesting activation patterns while minimizing the number of single iModulons found.

3. They claim that their approach will reveal response patterns of different gene groups as a function of time, which they did. They also claim that their ICA method can reveal details about TRN structure, cellular stresses associated with specific metals, and important relationships between co-expressed genes. However, there were no results demonstrating this claim about using this approach to identify gene-gene relationships.

We thank the reviewer for pointing out the necessity for clarity in presenting our demonstration of gene-gene relationships through our ICA method.

Although we don't directly show relationships between co-expressed genes, we show that this platform can generate hypotheses about why different genes might fall into different iModulons based on their relationships in metabolic pathways. The phenomenon we reported - the fast activation of the narZ (nitrate reductase) and the intermediate activation profile of nrfE (nitrite reductase) - indicates a sequential and interdependent activation pathway. The paragraph describing this finding is reported below:

"In particular, we were able to observe the temporal dynamics of the pathway activation since narZ (nitrate reductase) belonged to the fast activation iModulon and nrfE (nitrite reductase) was associated with the intermediate activation profile. The order of activation matches the order required for nitrogen respiration, with nitrate first being reduced to nitrite by narZ, then nitrite being reduced to ammonium by nrfE (wraage2001role). This mechanism follows the "just-in-time" transcription program in metabolic pathways that was previously suggested in the literature (zaslaver2004just)."

In order to avoid misunderstandings, we deleted this sentence and substituted with the following:

Line 38: "This analysis has the potential to shed light on TRN structure and to help hypothesize on the cellular responses to specific metals based on the temporal patterns of gene activations observed."

Response to Reviewer 2

Major concerns

- From the manuscript, it is not clear whether the authors applied the ICA workflow to the gene expression data previously obtained by Graham et al. Or whether they redid the experiments to obtain new data. This point must be clarified in the manuscript.

We thank the reviewer for pointing out this aspect that needed clarification. We would like to confirm that we applied the Independent Component Analysis (ICA) workflow to the gene expression data that had previously been obtained in the study conducted by Graham et al. We did not carry out new experiments to procure fresh data for this analysis; instead, we focused on applying a novel analytical approach to the existing dataset to derive new insights.

To avoid any confusion and to make this point crystal clear to all readers, we have updated the manuscript to expressly state this fact with the following sentence:

"In this study, we apply the ICA workflow to the gene expression dataset originally generated by Graham et al., thereby leveraging pre-existing data to derive unique insights into the

temporal patterns of activation of bacterial promoters in response to external heavy metal stress.“

- Lines 14-20: Dynamics was used with a pre-existing library of 2000 E. coli promoters. It is the most complete promoters' library available, but 2000 promoters are less than the half of total promoters in E. coli. The authors must assess how this incompleteness affects conclusions.

We appreciate the reviewer's observation regarding the utilization of a library consisting of 2000 E. coli promoters, which, while being the most comprehensive available, does indeed represent less than half of the total known promoters in E. coli.

In the context of our study, we believe that this library still offers a rich and diverse dataset that enables meaningful and substantial insights into E. coli gene expression dynamics. The library encompasses a broad array of promoter types, thereby permitting the representation of a wide spectrum of gene regulation mechanisms in E. coli, which we consider to be vital in drawing robust conclusions from our analysis.

However, we acknowledge that the incompleteness of the library poses a limitation to our study, as it might potentially exclude some rare or specific regulatory mechanisms. To address this, we have noted in the discussion section the necessity for further studies with an extended promoter library to validate and potentially enrich our findings. The text added is the following:

“While the library of 2000 E. coli promoters employed in this study represents the most comprehensive collection currently available and allows for substantial insights into gene expression dynamics, we acknowledge that it encompasses less than half of the total known promoters in E. coli. This limitation signifies that our analysis might not fully capture all the potential regulatory mechanisms at play, and we advocate for future studies to incorporate a more expansive promoter library to offer a more encompassing view of E. coli gene expression.”

- The methodology is hard to follow for a reader not familiar with the previous papers by the authors. There are gaps and imprecisions that the authors must address. This is key for providing a transparent methodology, thus enabling reproducibility.

We thank the reviewer for the valuable feedback. We understand the importance of presenting a methodology that is both transparent and accessible to readers with varying degrees of familiarity with our previous works.

In response to this, we have taken measures to enhance the detailedness of the methodological section, working to fill any existing gaps and clarify areas that may have previously harbored imprecisions. The added sentences are as follows:

- Line 360: “In our DBSCAN analysis, we used the following parameters: DISTANCE parameter, analogous to the epsilon parameter in traditional DBSCAN applications, was set to its default value of 0.1, determining the maximum distance between two points to be considered as in the same cluster. Meanwhile, the MIN_FRAC parameter, which specifies the minimum fraction of total data points required to form a valid cluster, was maintained at its default setting of 0.5. This implies that each cluster in our analysis contains at least 50\% of the total data points present in the dataset. The final independent components were defined as the centroid of each cluster in M, and the weightings were defined as the centroid of their corresponding weighting vectors in A. To ensure that the final components were consistent across multiple runs, we computed the clustered components 100 times, and selected the components that were identified in every run. The previously published code used to compute robust independent components is publicly available at github.com/SBRG/precise-db. We also added the scripts specifically used in this paper in the "precise_db" folder within the "ICA_dynamics" GitHub repository.”
- Line 325: “Detailed information on how the raw data was processed can be found in the original study (Graham et al, 2020). In summary, the data was first processed by subtracting the local background signal and then dividing the result by the background signal again in order to create a measure of the amplification of the signal over the background. Specifically, fluorescence values measured at a location outside the cell trap were subtracted from those obtained within the cell-containing regions. This method effectively removes signal noise not associated with the cells, thereby providing a more reliable representation of the cellular fluorescence signals”
- Line 388: “ Identification of primary heavy metal associated to each iModulon

We used a simple computational strategy that involved the analysis of the activation matrix data of each iModulon to associate them with the heavy metal that induced the highest aggregate expression over the six time points within the corresponding induction window. This approach allowed us to pinpoint the condition under which each iModulon exhibited the most pronounced change, thereby facilitating a data-driven classification. The script used to implement this classification can be found in the ICA_dynamics GitHub repository. “

- Line 396: Classification of iModulons based on activity coefficient pattern.

In our study, we employed a simple computational method to classify the iModulons based on the shape of their activity coefficients within the induction window of heavy metal enrichment. This method involved calculating the center of mass (CM) and identifying the highest peak (HP) within the specified window for each iModulon, wherein the center of mass represents the mean position of a given function. The iModulons

were categorized into three groups based on these calculations: Steady ($CM \leq 5$ and $HP \geq 4$), Intermediate ($CM \leq 4$ and $HP \leq 4$), and Fast ($CM \leq 4$ and $HP \leq 3$). iModulons that did not conform to these parameters were labeled as "other," with plans for further characterization on a case-by-case basis. We have made the corresponding code block available in the "iModulon_identification" script within our GitHub repository, and illustrated the calculated values and resultant classifications in Supplementary Figure S54. It's noteworthy that the methodology applied is tailored to the specific shapes encountered in our study. For future studies, especially where the shape of activity coefficients is unknown or difficult to categorize, we advocate for exploring more generalized approaches. Machine learning methodologies, particularly those centered around shape recognition, could offer a robust and adaptable solution for classifying transcriptional profiles regardless of the inherent patterns within the data.

Furthermore, we have established a GitHub repository where we have uploaded all the scripts and data used in our study. This repository serves as a comprehensive resource, offering a deep dive into the exact processes and analyses carried out, and allowing for straightforward reproduction of our results by providing open access to all the underlying materials and scripts used in our research.

We trust that with these augmentations to the manuscript and the availability of a detailed GitHub repository, we have significantly improved the transparency and reproducibility of our methodology, thereby addressing your concerns effectively.

Minor concerns

- Line 73: Figure 1 is missing.

We thank the reviewer for bringing this to our attention. Figure 1 has been added back to the manuscript.

- Lines 104-109: The authors classified the different types of activation patterns into 4 classes. A table or figure supporting these definitions will improve clarity.

We thank the reviewer for this suggestion. We added a table (below) that provides a qualitative characterization of the different activation states in the supplementary material.

Activation type	Activation pattern	Description
Fast		Promoters quickly activating early in the induction window
Intermediate		Promoters with peak activity in the middle of the induction window.
Steady		Promoters showing a continuous increase in activity throughout the induction window.
Partial Steady		Promoters active initially but repressed towards the end of the induction window.
Late		Promoters activate later in the induction window.

Additionally, we revisited our methodology to elucidate the classification of iModulons based on the shape of their activity coefficients within the induction window of heavy metal enrichment. This was executed by calculating the center of mass (CM) and identifying the highest peak (HP) within the specified window for each iModulon. The iModulons were then categorized into three distinct groups based on these calculations: Steady (CM ≤ 5 and HP ≥ 4), Intermediate (CM ≤ 4 and HP ≤ 4), and Fast (CM ≤ 4 and HP ≤ 3). Those iModulons not conforming to these parameters were labeled as "other," with further characterization planned on a case-by-case basis. The corresponding code block has been made available in the "iModulon_identification" script within our GitHub repository, and the calculated values along with the resultant classifications have been illustrated in Supplementary Figure S54.

- Lines 246-304: The results discussed in the section “Transcriptional response of the recovery post-induction” should use the same sub-headers structure (by iModulon) that the “Dynamic transcriptional response to heavy metal exposure” section to improve reading.

We thank the reviewer for this suggestion, we added the sub-headers to improve readability.